# Rab32/38-Dependent and -Independent Transport of Tyrosinase to Melanosomes in B16-F1 Melanoma Cells

**DOI:** 10.3390/ijms232214144

**Published:** 2022-11-16

**Authors:** Aya Nishizawa, Yuto Maruta, Mitsunori Fukuda

**Affiliations:** Laboratory of Membrane Trafficking Mechanisms, Department of Integrative Life Sciences, Graduate School of Life Sciences, Tohoku University, Aoba-ku, Sendai 980-8578, Miyagi, Japan

**Keywords:** endosome, Hps4, melanocyte, melanogenic enzyme, melanoma, melanosome, membrane traffic, Rab small GTPase, tyrosinase, tyrosinase-related protein 1 (Tyrp1)

## Abstract

B16-F1 melanoma cells have often been used as a model to investigate melanogenesis, but the evidence that melanosome biogenesis and transport occur by the same mechanisms in normal melanocytes and B16-F1 cells is insufficient. In this study, we established knockout B16-F1 cells for each of several key factors in melanogenesis, i.e., tyrosinase (Tyr), Hps4, Rab27A, and Rab32·Rab38 (Rab32/38), and then compared their phenotypes with the phenotypes of corresponding mutant mouse melanocyte cell lines, i.e., melan-c, melan-le, melan-ash, and Rab32-deficient melan-cht cells, respectively. The results showed that Tyr and Rab27A are also indispensable for melanin synthesis and peripheral melanosome distribution, respectively, in B16-F1 cells, but that Hps4 or its downstream targets Rab32/38 are not essential for Tyr transport in B16-F1 cells, suggesting the existence of a Rab32/38-independent Tyr transport mechanism in B16-F1 cells. We then performed comprehensive knockdown screening of Rab small GTPases and identified Rab10 and Rab24, previously uncharacterized Rabs in melanocytes, as being involved in Tyr transport under Rab32/38-null conditions. Our findings indicate a difference between the Tyr transport mechanism in melanocytes and B16-F1 cells in terms of Rab32/38-dependency and a limitation in regard to using melanoma cells as a model for melanocytes, especially when investigating the mechanism of endosomal Tyr transport.

## 1. Introduction

Melanocytes are specialized cells that contain pigmented organelles, called melanosomes, and play an important role in protecting the eyes and skin from ultraviolet damage [1]. Mammalian skin and hair pigmentation develops in a multi-step process. In the first step, melanosomes form from early endosomes in a stepwise fashion and mature around the nucleus by acquiring melanogenic enzymes via membrane traffic (from stage I melanosomes to stage IV melanosomes; the melanosome biogenesis step). In the second step, the mature melanosomes are transported to the cell periphery along the cytoskeleton (microtubules and actin filaments; the melanosome transport step). In the third step, the mature melanosomes are transferred to surrounding keratinocytes (or hair matrix cells) via melanocyte dendrites (melanosome transfer step). In the final step, the melanosomes in the keratinocytes are transported to the perinucleus to form supranuclear melanin caps (melanin cap formation step) [2,3,4,5].

Genetic defects in the above steps in the pigmentation process, specifically in the melanosome biogenesis and transport steps, are known to cause albinism, a group of hereditary diseases characterized by hypopigmentation of the hair and skin, e.g., oculocutaneous albinism, Griscelli syndrome, and Hermansky–Pudlak syndrome (HPS) [6,7,8]. In the past few decades, genetic and biochemical analyses of the gene products responsible for these diseases have revealed the existence of several key players in each step of the pigmentation process. In addition, the establishment of immortalized melanocyte cell lines from mouse albinism models, i.e., coat color mutant mice and functional studies of them at the cellular level have greatly accelerated our understanding of the molecular mechanisms responsible for melanosome biogenesis and transport [1,9]. For example, tyrosinase (Tyr)-deficient melan-c cells [10], Hps4-deficient melan-le cells [11], and Rab27A-deficient melan-ash cells [12] have been shown to be important resources for investigating melanosome biogenesis and transport. Melan-c and melan-le cells lack pigmented melanosomes because of a defect in melanin synthesis and in Tyr transport to melanosomes, respectively, whereas melan-ash cells are characterized by perinuclear melanosome aggregation because of a defect in actin-based melanosome transport. Despite their importance, however, these mutant cell lines and even wild-type (WT) melan-a cells [13] cannot be maintained forever because of the limited number of passages possible. This drawback makes it extremely difficult to apply recently developed genome-editing technologies, such as CRISPR/Cas9 technology [14,15], to these cells as a means of identifying new regulators of melanosome biogenesis and transport.

To overcome this drawback, we turned our attention to the mouse melanoma cell line B16-F1 cells [16] because they contain pigmented melanosomes and retain the capacity for unlimited proliferation. We thought it might be possible to use B16-F1 cells to establish knockout (KO) cell lines and search for new regulators involved in melanosome biogenesis and transport. However, since there was insufficient evidence to conclude that the same mechanisms of melanosome biogenesis and transport operate in both melanocytes and melanoma cells, it was important to generate KO B16-F1 cells for some of the previously established regulators of melanosome biogenesis or transport and compare their phenotypes with the phenotypes of corresponding mutant melanocyte cell lines.

In this study, we focused on three key regulators of melanogenesis in melanocytes, i.e., Tyr (a melanogenic enzyme required for “melanin synthesis”), Hps4 (a component of BLOC-3 complex required for “Tyr transport”), and Rab27A (a small GTPase required for “melanosome transport”) [5], and we established KO B16-F1 cells for each of them. The results showed that both Tyr and Rab27A are also required for the formation of pigmented melanosomes and peripheral melanosome distribution, respectively, in B16-F1 cells, but that Hps4 or its downstream proteins, Rab32/38 [17], are not essential for Tyr transport, suggesting the existence of a Rab32/38-independent Tyr transport mechanism in B16-F1 cells. We also identified new regulators of Tyr transport in B16-F1 cells by comprehensively investigating them for the presence of members of the Rab family of small GTPases, conserved membrane trafficking regulators in all eukaryotes [18,19,20,21]. Based on our findings, we discuss the meaning of the Rab32/38-dependent and -independent Tyr transport pathways in melanoma cells and suggest restricting the use of melanoma cells as a model for investigating the mechanism of melanogenic enzyme transport in melanocytes.

## 2. Results

### 2.1. Establishment and Analysis of Tyr-KO B16-F1 Cells

To determine whether the presence of Tyr is also essential for melanin synthesis to occur in B16-F1 cells, we generated *Tyr*-KO B16-F1 cells by using CRISPR/Cas9-mediated genome editing technology (Figure 1A) and confirmed the absence of Tyr in the KO cells by both genomic sequencing (Figure 1B) and immunoblotting with specific antibodies (Figure 1C, lane 2). Bright-field microscopy revealed that, in contrast to the parental WT B16-F1 cells (Figure 1A, top row), the *Tyr*-KO B16-F1 cells were completely transparent, the same as Tyr-deficient melan-c cells, a melanocyte cell line derived from *albino* mice [10] (Figure 1A, bottom row). A quantitative analysis demonstrated the total absence of mature melanosomes in the *Tyr*-KO B16-F1 cells (Figure 1D, left graph). Since re-expression of Tyr with C-terminal FLAG-tag (Tyr-FLAG), but not expression of FLAG-enhanced green fluorescent protein (EGFP) in *Tyr*-KO B16-F1 cells (Figure 1C, lanes 3 and 4) completely restored matured melanosomes (Figure 1D, right graph, and Figure 1E), we concluded that Tyr is essential for both melanin synthesis and the formation of black mature melanosomes to occur in B16-F1 cells.

### 2.2. Normal Melanin Content and Distribution of Melanogenic Enzymes in Hps4-KO B16-F1 Cells

Next, we turned our attention to Hps4, a protein involved in Tyr transport to melanosomes together with Hps1 in melanocytes [11,17,22] and generated *Hps4*-KO B16-F1 cells (Figure 2A). Although the *Hps4*-KO was successfully achieved (Figure 2B,C, lane 2), mature melanosomes were unexpectedly observed in the *Hps4*-KO B16-F1 cells, in contrast to Hps4-deficient melan-le cells, which exhibited a transparent phenotype, the same as melan-c cells did [11,22] (Figure 2A, far right panel). In a previous study, we showed that the signals of the melanogenic enzymes, Tyr and Tyr-related protein 1 (Tyrp1), were restricted to the perinuclear region (i.e., lack of peripheral signals; Figure 2D, right four panels) of melan-le cells, and that their protein levels were lower in melan-le cells [22]. In addition, all these phenotypes were completely rescued by re-expression of Hps4 in melan-le cells [22]. By contrast, however, normal peripheral distributions of Tyr and Tyrp1 were observed in the *Hps4*-KO B16-F1 cells (Figure 2D, middle left panels), the same as in WT B16-F1 cells and melan-a cells. Moreover, the protein expression levels of Tyr and Tyrp1 were unaffected in B16-F1 cells, irrespective of the presence or absence of Hps4 (Figure 2C). Consistent with the normal amounts and distributions of the melanogenic enzymes in *Hps4*-KO B16-F1 cells, they contained a normal amount of melanin, the same as in the WT B16-F1 cells (Figure 2E). These results, when taken together, suggested that, in contrast to cultured melanocytes, Hps4 is not required for melanogenic enzyme transport or the formation of black mature melanosomes in B16-F1 cells.

### 2.3. A Perinuclear Melanosome Aggregation Phenotype of Rab27A-KO B16-F1 Cells

Since there appeared to be differences between the regulation of melanogenic enzyme transport to melanosomes in B16-F1 cells and in cultured melanocytes (Figure 2), we also investigated whether melanosome transport in B16-F1 cells is regulated by the same mechanism as in cultured melanocytes. To do so, we focused on Rab27A, a small GTPase that is specifically involved in actin-based melanosome transport together with its effector melanophilin (also known as Slac2-a) and myosin-Va in melanocytes [23,24,25,26], and generated *Rab27A*-KO B16-F1 cells (Figure 3A–C, lane 2). As shown in Figure 3A, both *Rab27A*-KO B16-F1 cells and Rab27A-deficient melan-ash cells [12] exhibited perinuclear melanosome aggregation (right panels), in contrast to the peripheral distribution of melanosomes in WT B16-F1 cells and melan-a cells (left panels). The results of a quantitative analysis indicated that the perinuclear melanosome aggregation phenotype in *Rab27A*-KO B16-F1 cells (Figure 3D, left graph) was attributable to a lack of Rab27A protein because re-expression of Rab27A with EGFP, not expression of EGFP, in *Rab27A*-KO B16-F1 cells (Figure 3C, lanes 3 and 4) completely restored peripheral melanosome distribution (Figure 3D, right graph, and Figure 3E). Thus, Rab27A is also essential for actin-based melanosome transport and peripheral melanosome distribution to occur in B16-F1 cells.

### 2.4. Rab32/38-Independent Melanogenic Enzyme Transport to Melanosomes in B16-F1 Cells

Of the three proteins involved in skin pigmentation in mammals that we investigated in B16-F1 cells (Figure 1, Figure 2 and Figure 3), Hps4 is unlikely to be involved in melanogenic enzyme transport to melanosomes. Since Hps4, together with Hps1 (i.e., BLOC-3 complex), is known to function as an activator (i.e., guanine nucleotide exchange factor) of the small GTPases Rab32 and Rab38 (Rab32/38) [17,22], both of which redundantly regulate melanogenic enzyme transport in melanocytes [27,28,29,30], we hypothesized that Rab32/38 are activated in an Hps4-independent manner in *Hps4*-KO B16-F1 cells. To test this hypothesis, we biochemically compared the amount of active Rab32/38 in WT cells and *Hps4*-KO B16-F1 cells by performing GTP-Rab32/38 pull-down assays using a specific GTP-Rab32/38-binding domain (i.e., ankyrin repeat domain 1 [ANKR1]) of Varp [31]. As shown in Figure 4A–C, however, the amount of GTP-Rab32/38 in *Hps4*-KO B16-F1 cells was significantly reduced in comparison with the amount in WT cells (compare lanes 3 and 4, second panel from the top). These findings indicated that Rab32/38 are not sufficiently activated in *Hps4*-KO cells, suggesting that neither Rab32 nor Rab38 functions in melanogenic enzyme transport in B16-F1 cells, at least under Hps4-deficient conditions. To further determine whether Rab32/38 are also required for melanogenic enzyme transport in WT B16-F1 cells, we tried inhibiting their function by overexpressing either RUTBC1, a known Rab32/38 inactivator (i.e., GTPase-activating protein) [29,32], or the ANKR1 domain of Varp, which can trap active Rab32/38 in cultured melanocytes [31]. The results confirmed our previous findings that overexpression of EGFP-tagged RUTBC1 or ANKR1 in melan-a cells results in a strong reduction in Tyr signals (~80% reduction; i.e., ~20% of the Tyr signals in control EGFP-expressing cells) (Appendix A), which is presumably caused by the lysosomal degradation of the untransported Tyr protein. In sharp contrast, the reduction in Tyr signals was not so evident in RUTBC1- or ANKR1-overexpressing WT B16-F1 cells (Figure 4D–F): there was only an ~30% and 50% reduction in Tyr signals in RUTBC1-overexpressing and ANKR1-overexpressing WT B16-F1 cells, respectively (Figure 4E). These results, when taken together, indicated that Rab32/38 are partly responsible for melanogenic enzyme transport in WT B16-F1 cells and strongly suggested the existence of a Rab32/38-independent mechanism of melanogenic enzyme transport in B16-F1 cells.

To demonstrate the existence of the Rab32/38-independent mechanism, we also generated *Rab32/38*-double KO (*Rab32/38*-KO) B16-F1 cells (Figure 5A–C). As expected, neither the melanin content (Figure 5D) nor the distribution and amount of Tyr and Tyrp1 (Figure 5C,E) were affected by *Rab32/38*-KO in B16-F1 cells. These results were in sharp contrast to the findings in cultured melanocytes, where the ability to perform melanogenic enzyme transport is entirely dependent on Rab32/38: Rab32-depleted melan-cht cells (i.e., Rab32/38-deficient cells) exhibited Tyr signals clustering in the perinuclear region and a transparent phenotype (Figure 5F, bottom panels). We therefore concluded that the Rab32/38-independent mechanism should function during melanogenic enzyme transport in B16-F1 cells.

### 2.5. Screening for Other Candidate Rabs Involved in Melanogenic Enzyme Transport in B16-F1 Cells

To investigate the Rab32/38-independent mechanism of melanogenic enzyme transport in B16-F1 cells, we hypothesized that other previously uncharacterized Rab family members support melanogenic enzyme transport under Rab32/38-deficient conditions, and we tested our hypothesis by performing comprehensive knockdown (KD) screening in *Rab32/38*-KO B16-F1 cells by using specific small interfering RNAs (siRNAs) against mouse Rab1A–Rab43 that we had developed previously [33]. To circumvent functional compensation by closely related isoforms (e.g., Rab5A/B/C) [34], we knocked them down simultaneously. We immunostained Tyr and quantified the relative intensity of Tyr in each Rab-KD cell (Figure 6A,B). Initially, we selected the seven candidate Rabs (Rab5, Rab9, Rab10, Rab24, Rab28, Rab30, and Rab34) whose KD resulted in more than a 30% reduction in Tyr signals in comparison with the control siRNA-treated cells (Figure 6A, blue bars, and see also Appendix A for KD efficiency of siRNAs). Two of them, Rab5 and Rab9, have previously been reported to be involved in melanogenic enzyme transport in cultured melanocytes [29,35,36], thereby validating our screening method. We then knocked down these candidate Rabs in the WT B16-F1 cells and evaluated the KD effect on Tyr signals (Figure 6C). Under our experimental conditions, KD of Rab10, Rab24, or Rab34 slightly decreased the Tyr signals in WT B16-F1 cells, but the difference was not statistically significant (Figure 6C). To exclude a possible off-target effect of siRNAs, we used additional siRNA sites (#2) for Rab10, Rab24, and Rab34, and the results confirmed that KD of Rab10 or Rab24 significantly decreased Tyr signals both in WT cells and *Rab32/38*-KO B16-F1 cells (Figure 6D). By contrast, siRab34#2 had no significant effect on Tyr signals, although it clearly reduced endogenous Rab34 protein expression more efficiently than the siRab34#1 did (Figure 6E), suggesting that the effect of the siRab34#1 on Tyr signals is nonspecific (Figure 6E). The results of our comprehensive screening showed that Rab10 and Rab24 are likely to regulate Rab32/38-independent melanogenic enzyme transport in B16-F1 cells.

## 3. Discussion

It is generally known that cultured melanocytes, e.g., melan-a cells, are unable to grow forever and maintain their melanin-producing properties. Because of this limitation, it is difficult to apply genome-editing technologies such as the CRISPR/Cas9 technology to melanocytes as a means of searching for new regulators of melanosome biogenesis and transport by genome-wide screening. In the present study, we focused on B16-F1 melanoma cells and established several KO cells for key factors in melanogenesis, i.e., Tyr, Hps4, Rab27A, and Rab32/38, to determine whether the mechanisms of melanogenesis in melanocytes are retained in B16-F1 cells. The results showed that neither Hps4 nor its downstream targets Rab32/38 are required for Tyr transport to melanosomes in B16-F1 cells (Figure 2, Figure 4 and Figure 5), but that Tyr and Rab27A are indispensable for melanin synthesis (Figure 1) and actin-based melanosome transport (Figure 3), respectively. These findings indicated that the mechanisms of melanosome biogenesis, specifically the Tyr transport mechanism in melanocytes and B16-F1 cells are different and suggested the existence of a Rab32/38-independent Tyr transport mechanism in B16-F1 cells. We then succeeded in identifying Rab10 and Rab24 as alternative Rabs that are involved in Tyr transport under Rab32/38-deficient conditions by performing comprehensive KD screening of Rab GTPases (Figure 6).

Both Rab10 and Rab24 must also be partly responsible for Tyr transport in WT B16-F1 cells because the KD of each of them in the WT cells caused a 30% reduction in Tyr signals (Figure 6C,D). However, these findings cannot simply be interpreted as meaning that Rab10 and Rab24 can also support Tyr transport in melanocytes. These Rabs are unlikely to mediate Tyr transport in melanocytes, at least in humans, because skin and hair hypopigmentation are clearly observed even in HPS1 and HPS4 patients (i.e., Rab32/38-inactivated conditions) [8]. By contrast, the corresponding HPS model mice, *pale ear* (Hps1-deficient) and *light ear* (Hps4-deficient) exhibit pigmentation on their backs comparable to WT mice but develop hypopigmentation in several other parts of their bodies, including their ears, tail, and feet [11], suggesting the existence of a difference between the mechanisms of melanogenesis on the back skin and the ears [37]. We therefore speculate that an Hps1–Hps4 (or Rab32/38)-independent Tyr transport mechanism functions on the backs of *pale ear* and *light ear* mice and that Rab10 and/or Rab24 may be involved in this Tyr transport back-up mechanism in the absence of active Rab32/38 in mice. Further research will be needed to determine whether Rab10 and/or Rab24 are actually involved in Tyr transport in mouse back skin under Rab32/38-inactivated conditions.

How do Rab10 and Rab24 participate in Tyr transport? This is an important question that needs to be addressed first in our next study. Although nothing is known about the function of Rab10 in melanocytes, Rab10 has previously been reported to localize to several organelles, including the Golgi apparatus [38] and tubular recycling endosomes [39,40], both of which are important organelles for Tyr transport to melanosomes [5,41] in other cell types. We especially noted that Rab10 has been found to regulate tubular endosome formation in HeLa cells through interaction with KIF13A and KIF13B [40] and that depletion of KIF13A in melanocytes inhibits melanosome biogenesis (e.g., inhibition of Tyrp1 transport) [41,42]. Thus, it is tempting to speculate that Rab10 promotes Tyr transport to melanosomes via tubular endosomes through interaction with KIF13A, especially under BLOC-3-deficient (or Rab32/38-inactivated) conditions.

No research has ever been conducted on the role of Rab24 in melanocytes. In other cell types, Rab24 has been shown to localize to several organelles, including late endosomes, and to be involved in endo-lysosomal degradation [43] and autophagy [44,45]. Thus, Rab24 may also be involved in the endosomal transport of Tyr. Alternatively, since several autophagic regulators, e.g., ULK1 and WIPI1, have been reported to regulate melanosome biogenesis [46,47], Rab24 may be involved in melanosome biogenesis through the regulation of certain autophagic regulators. Further extensive research will be necessary to reveal the precise molecular mechanism by which Rab10 and Rab24 regulate Tyr transport in B16-F1 cells and to determine the functional relationship between Rab10 and Rab24 in Tyr transport.

## 4. Materials and Methods

### 4.1. Materials

The oligonucleotides, plasmids, and antibodies used in this study are summarized in Appendix A. Unless otherwise specified, all other general reagents used in this study were analytical grade or the highest grade commercially available.

### 4.2. Cell Cultures and Transfections

The black-mouse-derived immortal melanocyte cell line melan-a [13] (obtained from the Wellcome Trust’s Functional Genomics Cell Bank at St George’s, University of London), melan-c (Tyr-deficient) [10], melan-le (Hps4-deficient) [11], melan-ash (Rab27A-deficient) [12], and melan-cht (Rab38-deficient) [28] were cultured as described previously [22,48] (see also the Wellcome Trust Functional Genomics Cell Bank homepage; https://www.sgul.ac.uk/about/our-institutes/molecular-and-clinical-sciences/research-sections/cell-biology-research-section/genomics-cell-bank). The B16-F1 melanoma cells (obtained from the American Type Culture Collection, Manassas, VA, USA) and COS-7 cells were cultured at 37 °C under 5% CO_2_ in D-MEM (Fujifilm Wako Pure Chemical, Osaka, Japan) containing 10% fetal bovine serum, 100 units/mL penicillin G, and 100 µg/mL streptomycin. For the KD experiments, B16-F1 cells were transfected with siRNAs (final 100 nM) by using RNAiMAX (Thermo Fisher Scientific, Waltham, MA, USA) according to the manufacturer’s instructions, and then cultured for 72 h. The Melan-cht cells were similarly transfected with *Rab32* siRNA (final 100 nM), and after 48 h culture were transfected with the same siRNA again and cultured for an additional 120 h. For the immunofluorescence analysis, the B16-F1 cells and melanocyte cell lines were transfected with plasmid DNAs by using Lipofectamine 2000 (Thermo Fisher Scientific) according to the manufacturer’s instructions, and then incubated for 24–72 h.

### 4.3. Immunofluorescence Analysis

The B16-F1 cells and melanocyte cell lines were fixed with 4% paraformaldehyde, permeabilized with 0.05% saponin, stained with specific primary antibodies (antibody dilutions are summarized in Appendix A), and then visualized with Alexa Fluor 488/555-conjugated secondary antibodies. The stained cells were examined for fluorescence with a confocal fluorescence microscope (FluoView 1000-D, Evident/Olympus, Tokyo, Japan) through an objective lens (×60 magnification, N.A. 1.40, Evident/Olympus) and with the FluoView software (version 4.1a, Evident/Olympus). The fluorescence images and their corresponding bright-field images were captured at random with the confocal microscope and quantified with the ImageJ software (version 1.52i; National Institutes of Health, Bethesda, MD, USA).

### 4.4. Establishment of B16-F1 KO Cells

The CRISPR/Cas9-mediated KO of B16-F1 cells was performed essentially as described previously [34]. The specific primers for the single-guide RNAs (sgRNAs) used in this study are summarized in Appendix A. The KO of target genes was confirmed by both sequencing genomic PCR products (see also genomic PCR primers in Appendix A) and immunoblotting.

### 4.5. GTP-Rab32/38 Pull-Down Assays in B16-F1 Cells

Glutathione-Sepharose beads (GE Healthcare, Chicago, IL, USA) coupled with T7- glutathione *S*-transferase (GST) or T7-GST-ANKR1 were prepared as described previously [22]. The B16-F1 cells (WT or *Hps4*-KO) were lysed with a lysis buffer (50 mM HEPES-KOH pH 7.2, 150 mM NaCl, 1 mM MgCl_2_, and 1% Triton X-100 supplemented with a complete EDTA-free protease inhibitor mixture [Roche, Basel, Switzerland]) and centrifuged at 20,400× *g* for 10 min at 4 °C. The supernatant obtained was incubated with the T7-GST-ANKR1 (or T7-GST)-immobilized beads for 1 h at 4 °C. The proteins bound to the beads were analyzed by 10% SDS-PAGE followed by immunoblotting with the appropriate antibodies (antibody dilutions are summarized in Appendix A), as indicated in each figure. The immunoreactive bands were visualized by enhanced chemiluminescence as described below.

### 4.6. Immunoblotting

Protein samples were subjected to 10% or 12.5% SDS-PAGE and transferred to a polyvinylidene difluoride membrane (Merck Millipore, Burlington, MA, USA) by electroblotting. The blots were blocked with 1% or 3% skim milk and incubated at room temperature with primary antibodies (antibody dilutions are summarized in Appendix A) for 1 h and then with appropriate secondary antibodies conjugated with horseradish peroxidase. Immunoreactive bands were detected by enhanced chemiluminescence (GE Healthcare) using a chemiluminescence imager (ChemiDoc Touch; Bio-Rad, Hercules, CA, USA). The blots shown in the figures are representative of the results obtained in two or three independent experiments.

### 4.7. Melanin Assays

The Melanin assays were performed essentially as described previously [31]. In brief, the B16-F1 cells (WT and KO) were solubilized in 50 mM HEPES-KOH, pH 7.2, 150 mM NaCl, and 1% Triton X-100. The pigment was then pelleted by centrifugation at 20,400× *g* for 10 min at 4 °C, and the pellet was dissolved in 2N NaOH/20% DMSO for 30 min at 100 °C. Melanin content was measured as optical density at 490 nm with a microplate reader (model 680XR, Bio-Rad; Figure 2E) or Victor Nivo Multimode microplate reader (PerkinElmer, Waltham, MA, USA; Figure 5D) and normalized to total protein content.

### 4.8. Reverse-Transcription (RT)-PCR

The total RNA of the B16-F1 cells that had been transfected with *Rab* siRNAs (for Rab5A/B/C, Rab9A/B, Rab10, Rab24, Rab28, Rab30, and Rab34) or control siRNA were lysed with TRI Reagent (Sigma-Aldrich, St. Louis, MO, USA). RT was performed by using the purified RNA and ReverTra Ace^®^ (Toyobo, Osaka, Japan), and cDNAs were amplified by performing PCR with specific oligonucleotides (summarized in Appendix A) according to the manufacturer’s instructions.

### 4.9. Statistical Analysis

The statistical analysis was performed by using Tukey’s test for differences between more than two samples (Figure 2E and Figure 5D), Dunnett’s test for differences between more than two samples (Figure 4E and Figure 6C,D), and Student’s unpaired *t*-test (for differences between two samples in Figure 1D, Figure 3D, and Figure 4C). In all of the tests, *p* values < 0.05 were considered statistically significant.

## 5. Conclusions

In summary, in the study reported here we have demonstrated a difference between the Tyr transport mechanism in terms of Rab32/38-dependency in melanocytes and B16-F1 cells, and for the first time identified Rab10 and Rab24 as possible players in Tyr transport. Although melanoma cells, including B16-F1 cells, have often been used as a model to investigate melanogenesis in the fields of cell biology and cosmetics, our findings suggest a limitation to the use of melanoma cells as a model of melanocytes, that is, that melanoma cells would not be an appropriate tool for investigating the mechanism of melanogenic enzyme transport or searching for new cosmetics or drugs that inhibit melanogenic enzyme transport.

## Figures and Tables

**Figure 1 ijms-23-14144-f001:**
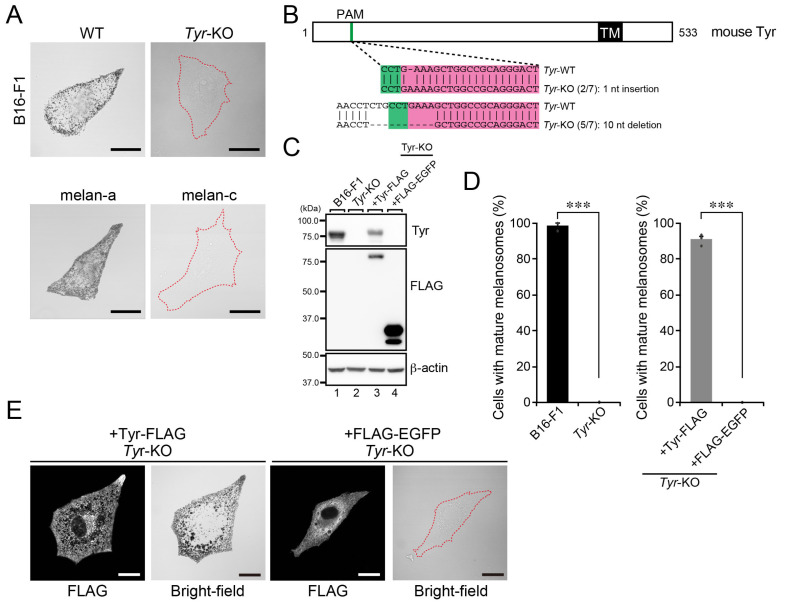
Tyr is essential for the formation of black-pigmented mature melanosomes. (**A**) Representative bright-field images of WT B16-F1 cells, *Tyr*-KO B16-F1 cells, melan-a cells, and Tyr-deficient melan-c cells. Cells devoid of black-pigmented melanosomes are outlined with broken red lines. Scale bars, 20 μm. (**B**) Domain organization of mouse Tyr protein and genomic mutations in *Tyr*-KO cells. The Cas9 target and PAM (protospacer adjacent motif) sequences are highlighted in magenta and green, respectively. Genomic PCR products containing the target site were subcloned into the pGEM-T Easy vector, and then seven colonies (the denominators of the fractions in the parentheses) were picked up at random, and their inserts were sequenced. The numerators of the fraction in parentheses are the numbers of colonies obtained by sequencing the subcloned PCR products. TM, transmembrane domain. (**C**) Expression of Tyr in WT cells and *Tyr*-KO B16-F1 cells. Lysates of the cells indicated were analyzed by immunoblotting with antibodies against Tyr (top), FLAG tag (middle), and β-actin (bottom). (**D**) Quantification of the numbers of WT cells, *Tyr*-KO cells, *Tyr*-KO cells expressing Tyr-FLAG, and *Tyr*-KO cells expressing FLAG-EGFP containing black-pigmented melanosomes shown in (**A**,**E**). The graphs show the means and SEM of the data obtained in three independent experiments (*n* > 26 cells in each experiment). ***, *p* < 0.001 (Student’s unpaired *t*-test). (**E**) Representative bright-field images of *Tyr*-KO cells expressing Tyr-FLAG or FLAG-EGFP. FLAG-tagged proteins were detected with an anti-FLAG tag antibody. The FLAG-EGFP-expressing cell is outlined with a broken red line. Scale bars, 20 μm.

**Figure 2 ijms-23-14144-f002:**
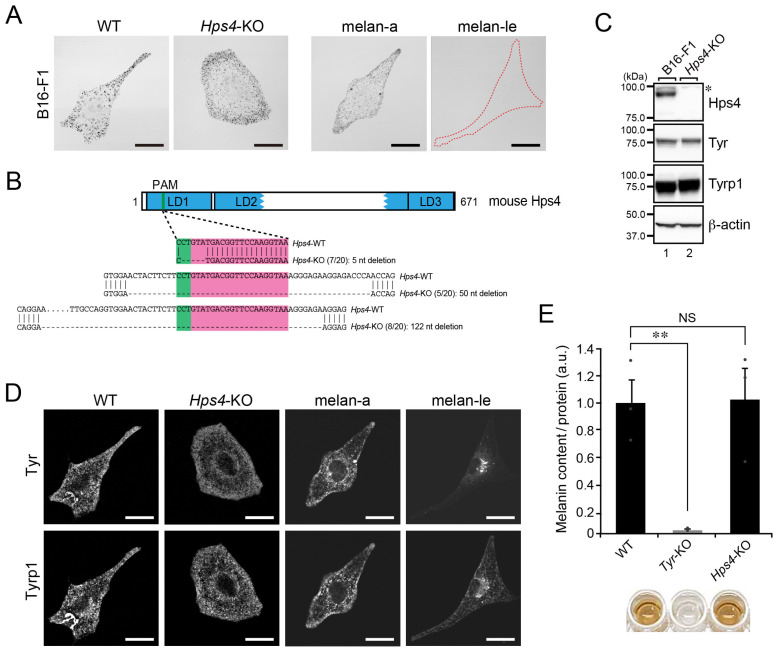
Black-pigmented melanosomes are formed in B16-F1 cells, even in the absence of Hps4. (**A**) Representative bright-field images of WT B16-F1 cells, *Hps4*-KO B16-F1 cells, melan-a cells, and Hps4-deficient melan-le cells. The melan-le cell devoid of black-pigmented melanosomes is outlined with a broken red line. Scale bars, 20 μm. (**B**) Domain organization of the mouse Hps4 protein and genomic mutations in *Hps4*-KO cells. The Cas9 target and PAM sequences are highlighted in magenta and green, respectively. Genomic PCR products containing the target site were subcloned into the pGEM-T Easy vector, and then 20 colonies (denominators of the fractions in parentheses) were picked up at random and their inserts were sequenced. The numerators of the fractions in the parentheses are the numbers of colonies obtained by sequencing the subcloned PCR products. LD1–3, longin domain 1–3. (**C**) Expression of Hps4 in WT cells and *Hps4*-KO B16-F1 cells as determined by immunoblotting with the antibodies indicated. The asterisk indicates a nonspecific band. (**D**) Representative images of Tyr and Tyrp1 in WT B16-F1 cells, *Hps4*-KO B16-F1 cells, melan-a cells, and Hps4-deficient melan-le cells. Their bright-field images are shown in (**A**). Scale bars, 20 μm. (**E**) Quantification of the melanin content (i.e., optical density at 490 nm) in WT cells, *Tyr*-KO cells, and *Hps4*-KO B16-F1 cells. The graph shows the means and SEM of the data obtained in three independent experiments. **, *p* < 0.01; NS, not significant (Tukey’s test).

**Figure 3 ijms-23-14144-f003:**
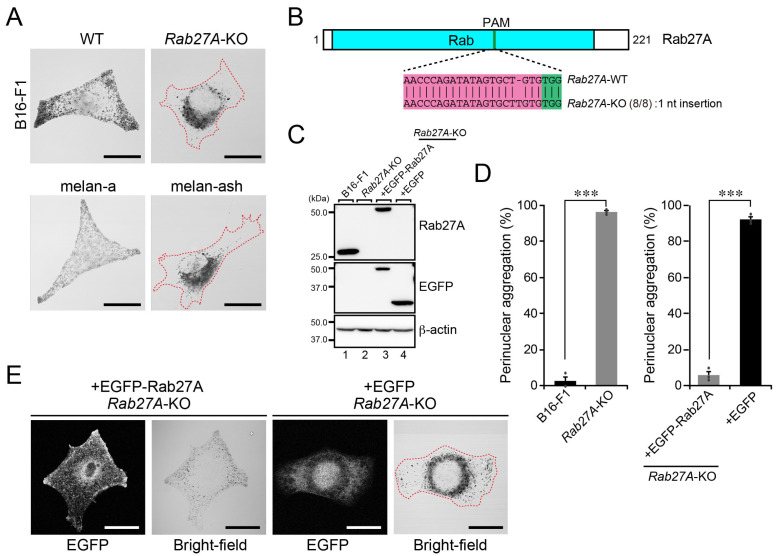
Rab27A is essential for the peripheral distribution of melanosomes. (**A**) Representative bright-field images of WT B16-F1 cells, *Rab27A*-KO B16-F1 cells, melan-a cells, and Rab27A-deficient melan-ash cells. Cells showing perinuclear melanosome aggregation are outlined with broken red lines. Scale bars, 20 μm. (**B**) Domain organization of the mouse Rab27A protein and genomic mutations in *Rab27A*-KO cells. The Cas9 target and PAM sequences are highlighted in magenta and green, respectively. Genomic PCR products containing the target site were subcloned into the pGEM-T Easy vector. Then, eight colonies (denominators of the fractions in parentheses) were picked up at random and their inserts were sequenced. (**C**) Expression of Rab27A in WT cells and *Rab27A*-KO B16-F1 cells. Lysates of the cells indicated were analyzed by immunoblotting with antibodies against Rab27A (top), GFP (middle), and β-actin (bottom). (**D**) Quantification of the numbers of WT cells, *Rab27A*-KO cells, *Rab27A*-KO cells expressing EGFP-Rab27A, and *Rab27A*-KO cells expressing EGFP with perinuclear melanosome aggregation shown in (**A**,**E**). The graphs show the means and SEM of the data obtained in three independent experiments (*n* > 24 cells in each experiment). ***, *p* < 0.001 (Student’s unpaired *t*-test). (**E**) Representative bright-field images of *Rab27A*-KO cells expressing EGFP-Rab27A or EGFP alone. The EGFP-expressing cell is outlined with a broken red line. Scale bars, 20 μm.

**Figure 4 ijms-23-14144-f004:**
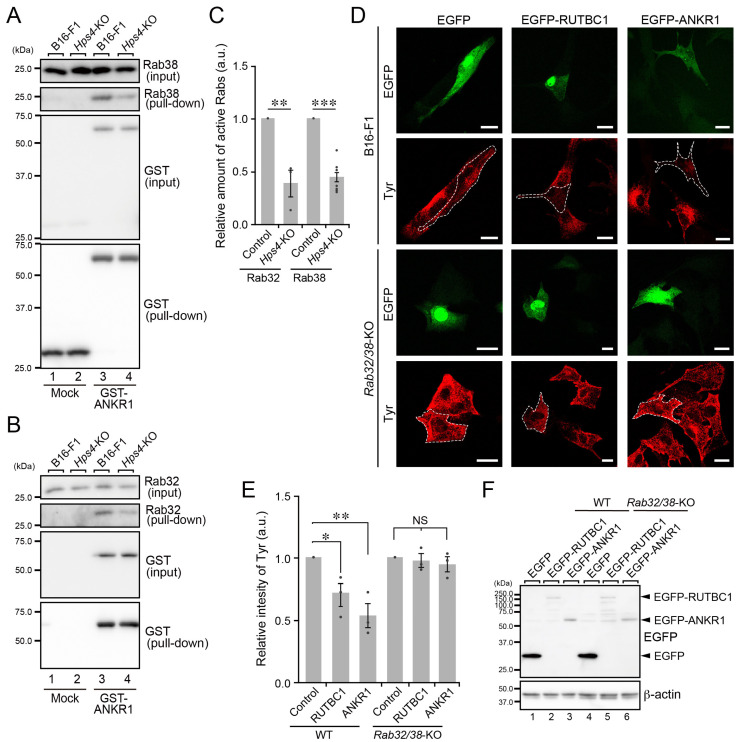
Decreased activation of Rab32/38 in *Hps4*-KO B16-F1 cells and Rab32/38-independent Tyr transport in *Rab32/38*-KO B16-F1 cells. (**A**) The amount of active Rab38 as determined by GTP-Rab38 pull-down assays, which were performed using the ANKR1 domain of Varp, i.e., Rab32/38 effector domain [22]. Beads coupled with GST alone (control) or GST-Varp-ANKR1 were incubated with lysates of WT cells or *Hps4*-KO B16-F1 cells, and the proteins bound to the beads were analyzed by immunoblotting with the antibodies indicated. (**B**) The amount of active Rab32 as determined by the GTP-Rab32 pull-down assays, which were performed using GST-Varp-ANKR1 essentially as described in (**A**). (**C**) The intensity of the active GTP-Rab32 bands (second panel in (**B**); *n* = 3) and GST-Rab38 bands (second panel in (**A**); *n* = 5) normalized by the total number of Rab32 and Rab38 bands (top panel in (**A**,**B**)), respectively, were quantified and analyzed statistically (**, *p* < 0.01; ***, *p* < 0.001; Student’s unpaired *t*-test). (**D**) Representative Tyr images (red) of WT cells and *Rab32/38*-KO B16-F1 cells expressing either EGFP alone, EGFP-RUTBC1 [29], or EGFP-ANKR1 [31]. The EGFP-expressing cells (green) are outlined with broken white lines. Scale bars, 20 μm. (**E**) Relative intensity of Tyr in the WT cells and *Rab32/38*-KO B16-F1 cells shown in (**D**). The graph shows the means and SEM of the data obtained in three independent experiments (*n* > 10 cells in each experiment). *, *p* < 0.05; **, *p* < 0.01; NS, not significant (Dunnett’s test). (**F**) Expression of EGFP-tagged proteins in WT cells and *Rab32/38*-KO B16-F1 cells. Lysates of the cells indicated were analyzed by immunoblotting with antibodies against GFP (top) and β-actin (bottom).

**Figure 5 ijms-23-14144-f005:**
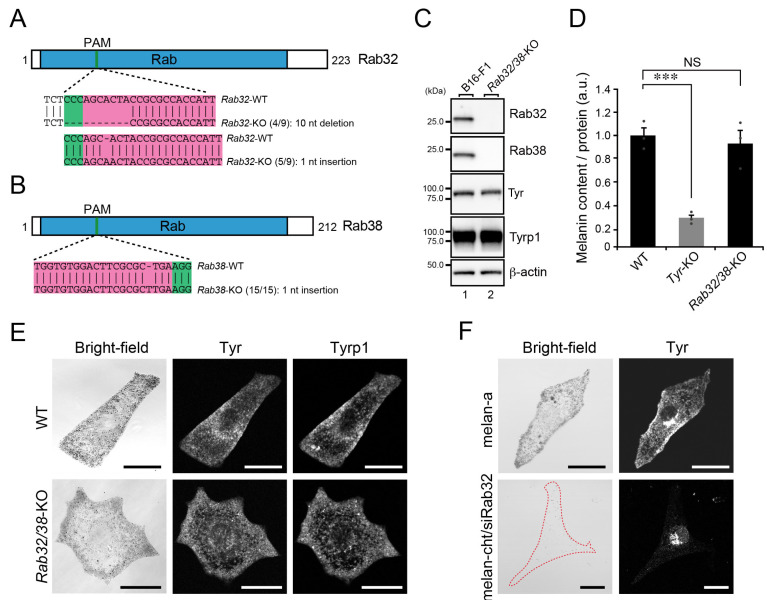
Black-pigmented melanosomes formed normally in *Rab32*/*Rab38*-KO B16-F1 cells. (**A**,**B**) Domain organization of the mouse Rab32 and Rab38 proteins and genomic mutations in *Rab32*/*38*-KO cells. The Cas9 target and PAM sequences are highlighted in magenta and green, respectively. Genomic PCR products containing the target sites were subcloned into the pGEM-T Easy vector. Then, nine colonies for Rab32 and 15 colonies for Rab38 (denominators of the fractions in parentheses) were picked up at random and their inserts were sequenced. The numerators of the fractions in the parentheses are the numbers of colonies obtained by sequencing the subcloned PCR products. (**C**) Expression of Rab32, Rab38, Tyr, and Tyrp1 in WT cells and *Rab32/38*-KO B16-F1 cells as determined by immunoblotting with the antibodies indicated. (**D**) Quantification of the melanin content (i.e., optical density at 490 nm) in WT cells, *Tyr*-KO cells, and *Rab32/38*-KO B16-F1 cells. The graph shows the means and SEM of the data obtained in three independent experiments. ***, *p* < 0.001; NS, not significant (Tukey’s test). (**E**) Representative images of Tyr and Tyrp1 in WT cells and *Rab32/38*-KO B16-F1 cells, and their corresponding bright-field images. Scale bars, 20 μm. (**F**) Representative images of Tyr in melan-a cells and Rab32-deficient melan-cht cells, and their corresponding bright-field images. Scale bars, 20 μm.

**Figure 6 ijms-23-14144-f006:**
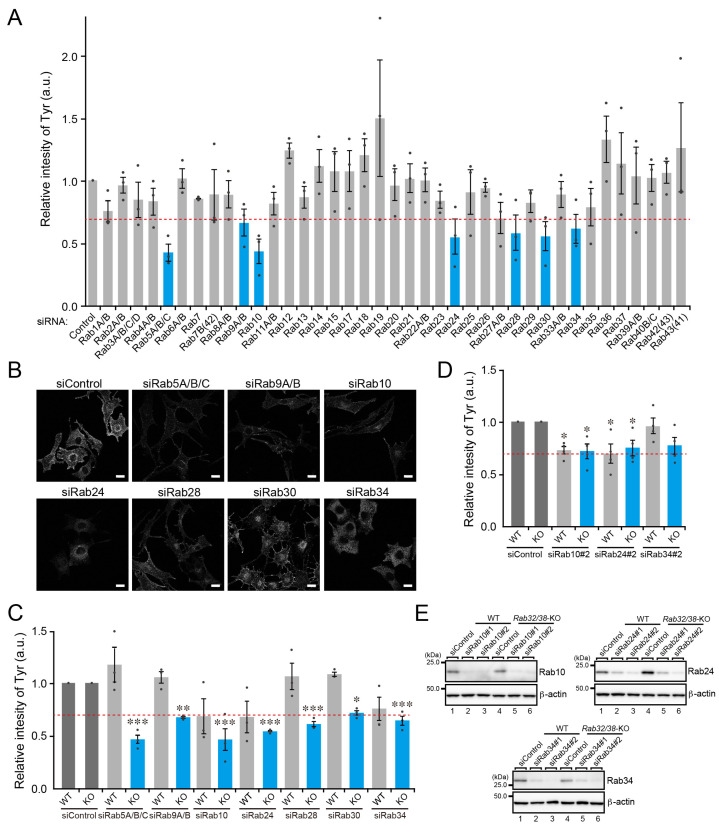
Comprehensive screening for Rabs whose KD resulted in a reduction in Tyr signals in *Rab32/38*-KO B16-F1 cells. (**A**) Relative intensity of Tyr in *Rab32/38*-KO cells transfected with control siRNA or siRNA against each Rab subfamily was determined by a quantitative immunofluorescence analysis. Error bars indicate the SEM of the data obtained in three independent experiments (*n* > 10 cells in each experiment). The broken red line indicates a 30% reduction in Tyr signals. The blue bars represent the seven candidate Rabs whose KD resulted in a more than 30% reduction in Tyr signals in comparison with the control siRNA. KD of these candidate Rabs was confirmed by RT-PCR analysis (see Appendix A). (**B**) Typical images of Tyr in the candidate Rab-KD cells shown in (**A**). Scale bars, 20 μm. (**C**) Relative intensity of Tyr in WT cells (light gray bars) and *Rab32/38*-KO B16-F1 cells (blue bars) transfected with control siRNA or siRNA against each candidate Rab shown in (**A**), as determined by quantitative immunofluorescence analysis. Error bars indicate the SEM of the data obtained in three independent experiments (*n* > 10 cells for each experiment). The broken red line indicates the 30% reduction of Tyr signals. *, *p* < 0.05; **, *p* < 0.01; ***, *p* < 0.001 in comparison with the results obtained with the control siRNA (Dunnett’s test). (**D**) Relative intensity of Tyr in WT cells (light gray bars) and *Rab32/38*-KO B16-F1 cells (blue bars) transfected with control siRNA or siRNA#2 against Rab10, Rab24, and Rab34. Error bars indicate the SEM of the data obtained in four independent experiments (*n* > 20 cells for each experiment). The broken red line indicates the 30% reduction of Tyr signals. *, *p* < 0.05 in comparison with the results obtained with the control siRNA (Dunnett’s test). (**E**) KD efficiency of siRNAs toward Rab10, Rab24, and Rab30 as revealed by immunoblotting with the specific antibodies indicated.

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
