# Peer review of "Rab32/38-Dependent and -Independent Transport of Tyrosinase to Melanosomes in B16-F1 Melanoma Cells"

_ijms, 2022, doi:10.3390/ijms232214144_

Round 1

Reviewer 1 Report

Nishizawa et al. “Rab32/38-dependent and -independent transport of tyrosinase to melanosomes in B16-F1 melanoma cells”

In this manuscript, the authors present the results of experiments designed to assess whether B16-F1 melanoma cells are good models of melanogenesis in normal physiology, which can be used to further define the pathway.  The experimental approach was to determine the effect of knockout of proteins known to be involved in melanogenesis in nontransformed melanocytes. The effects of tyrosinase and Rab27a were consistent with known function in the pathway, but HSP4, Rab32 and Rab38 knockout did not affect melanogenesis in these cells.  A screen of knockdown of multiple members of the Rab family revealed a potential function of Rab10 and Rab24, replacing the function of bloc-3, Rab32 and Rab38.  The work provides valuable information for the field, but could be strengthened in some respects prior to publication.

1.      There is no indication of reproducibility of the Rab pull downs (unless I missed it), e.g. number of times the experiments were performed.  The manuscript would be strengthened by either showing westerns from three experiments or quantifying and showing the data summary in a Bar graph.

2.     The critical finding of the paper is that Rab28, Rab34 are dispensable and Rab10 and Rab24 may assume their role, or drive a separate pathway.  Because this result is important for establishing the difference between melanocytes and melanoma, it should be further tested.  Two siRNA do not necessarily exclude an off target effect.  A rescue would be ideal, but a third siRNA would help.  Also, further characterization would be valuable.  What happens when both Rab10 and Rab24 are knocked down simultaneously? Can you get a complete or partial rescue of the double knock down with Rab10? Rab24?

Author Response

[Response to the reviewer #1]

In this manuscript, the authors present the results of experiments designed to assess whether B16-F1 melanoma cells are good models of melanogenesis in normal physiology, which can be used to further define the pathway.  The experimental approach was to determine the effect of knockout of proteins known to be involved in melanogenesis in nontransformed melanocytes.  The effects of tyrosinase and Rab27a were consistent with known function in the pathway, but HSP4, Rab32 and Rab38 knockout did not affect melanogenesis in these cells.  A screen of knockdown of multiple members of the Rab family revealed a potential function of Rab10 and Rab24, replacing the function of bloc-3, Rab32 and Rab38.  The work provides valuable information for the field, but could be strengthened in some respects prior to publication.

              We thank the reviewer’s very positive comments and helpful suggestions.

Specific Comments:

  1. There is no indication of reproducibility of the Rab pull downs (unless I missed it), e.g. number of times the experiments were performed. The manuscript would be strengthened by either showing westerns from three experiments or quantifying and showing the data summary in a Bar graph.

              We have performed active Rab pull-down assays more than three times and obtained similar results.  As suggested by the reviewer, the intensity of active GTP-Rab32/38 bands normalized by total Rab32/38 bands were quantified and analyzed statistically (see new Figure 4C; **, p < 0.01; ***, p < 0.001; Student’s unpaired t-test).

  1. The critical finding of the paper is that Rab28, Rab34 are dispensable and Rab10 and Rab24 may assume their role, or drive a separate pathway. Because this result is important for establishing the difference between melanocytes and melanoma, it should be further tested. Two siRNA do not necessarily exclude an off target effect.  A rescue would be ideal, but a third siRNA would help.  Also, further characterization would be valuable.  What happens when both Rab10 and Rab24 are knocked down simultaneously?  Can you get a complete or partial rescue of the double knock down with Rab10? Rab24?

              Although we usually use two independent siRNAs to exclude an off target effect, we have further tested the third site of siRNAs for Rab10 and Rab24.  As shown in Figure S3 (for reviewers only), both siRab10#3 and siRab24#3 significantly reduced Tyr signals in WT/DKO B16-F1 cells, confirming our previous findings.  We also think that double KD of Rab10 and Rab24 would be interesting, but we would like to investigate the effect of double KD on Tyr signals in more detail in our future study (page 11, lines 371-372).

Reviewer 2 Report

The manuscript by Nishizawa et al found that there is an HPS4-independent pathway of melanosome biogenesis in B16-F1 cells differently from melan-le cells, and further found that Rab10 and Rab24 are the novel proteins to be involved in this pathway. As the current study found the unexpected pathway of HPS4-independent pathway that might be in the back of mice, this study represents significant advancement of the field. They can further improve the manuscripts as described below.

Major Comments

1. In Figure 2, the authors showed HPS4-KO B16-F1 cells have melanosomes, but melan-le cells do not. To confirm the depletion of melanosomes in melan-le cells are truly due to HPS4 depletion but not some off-target effects, rescue experiments such as overexpression of HPS4 to melan-le cells to see if melanosomes are seen or not, would be needed. Or at least, the authors should describe the reference that the melanosome depletion in melan-le cells is really caused by HPS4 but not by off-target effects from past publication. 

Minor comments

2. In Figure 4C the authors showed the decrease of tyrosinase intensity in B16-F1 WT but not in Rab32/38 B16-F1 cells and quantified the signal intensity in Figure 4D. In Figure 4C, ANKR1 expressing cells showed significant reduction of Tyr but the text described “only an ~50% reduction in Tyr signals in ANKR1-overxpressing WT B16-F1 cells~~”. Also in RUTBC1 expressing cells, ~80% and ~20% reduction in Figure 4C but there is no significant reduction in Figure 4D. These data and text would confuse readers. Is it possible that Rab32/38 inhibit the tyrosinase pathway in B16-F1 cells? If so, it is not so strange that overexpression of ANKR1 or RUTBC1 inhibits tyrosinase transport to melanosomes and redirect it to lysosomes, but no effect in tyrosinase and melanosomes by depletion of Rab32/28 in B16-F1 cells. The authors should review the data again and correct the text. In addition, although how Rab32/28 are involved in the melanosome biogenesis in B16-F1 cells is unclear, the authors should discuss about some possibilities in discussion. 

3. The authors described the distribution of Tyr and Tyrp1 are not affected in Rab32/38-DKO B16-F1 cells (line266,p7). But Tyr and Tyrp1 seem to be more localized near the Golgi, like melan-cht/siRab32 cells in Figure 5E. is it really no change of Tyr or Tyrp1 distribution in Rab32/38 B16-F1 cells? If there is some quantification of Tyr or Tyrp1 distribution in WT and Rab32/28-depleted B16-F1 cells, it is better to be noted in text. In addition, it is neccessary to describe the relationship between the change of distribution of Tyr and melanogenesis, to clarify which distribution of Tyr is important for melanogenesis.  

4. in p2, line85, “insvestingating” should be investigating.

5. In Figure 1B, is it 10 nt deletion instead of “insertion”?

6. Figure 3 legend should be in line 190, but not 200 in p5. 

7. In Figure S1, “E” should be C. 

Author Response

[Response to the reviewer #2] 

The manuscript by Nishizawa et al found that there is an HPS4-independent pathway of melanosome biogenesis in B16-F1 cells differently from melan-le cells, and further found that Rab10 and Rab24 are the novel proteins to be involved in this pathway.  As the current study found the unexpected pathway of HPS4-independent pathway that might be in the back of mice, this study represents significant advancement of the field.  They can further improve the manuscripts as described below.

              We thank the reviewer’s careful assessment of our manuscript and helpful suggestions.

Major comments:

  1. In Figure 2, the authors showed HPS4-KO B16-F1 cells have melanosomes, but melan-le cells do not. To confirm the depletion of melanosomes in melan-le cells are truly due to HPS4 depletion but not some off-target effects, rescue experiments such as overexpression of HPS4 to melan-le cells to see if melanosomes are seen or not, would be needed. Or at least, the authors should describe the reference that the melanosome depletion in melan-le cells is really caused by HPS4 but not by off-target effects from past publication.

              We agreed that it is important whether re-expression of Hps4 in melan-le cells restores black-pigmented melanosomes.  Actually, however, we have already reported that Hps4-expressing melan-le cells contain normal mature melanosomes (see Fig. 8A in ref. 22; J. Biol. Chem. 2019, 294, 6912–6922).  We have included this information in the revised manuscript (page 4, lines 133-134).

Minor comments:

  1. In Figure 4C the authors showed the decrease of tyrosinase intensity in B16-F1 WT but not in Rab32/38 B16-F1 cells and quantified the signal intensity in Figure 4D. In Figure 4C, ANKR1 expressing cells showed significant reduction of Tyr but the text described “only an ~50% reduction in Tyr signals in ANKR1-overxpressing WT B16-F1 cells”. Also in RUTBC1 expressing cells, ~80% and ~20% reduction in Figure 4C but there is no significant reduction in Figure 4D.  These data and text would confuse readers.  Is it possible that Rab32/38 inhibit the tyrosinase pathway in B16-F1 cells? If so, it is not so strange that overexpression of ANKR1 or RUTBC1 inhibits tyrosinase transport to melanosomes and redirect it to lysosomes, but no effect in tyrosinase and melanosomes by depletion of Rab32/28 in B16-F1 cells.  The authors should review the data again and correct the text.  In addition, although how Rab32/28 are involved in the melanosome biogenesis in B16-F1 cells is unclear, the authors should discuss about some possibilities in discussion.

              We apologize for this confusion.  In the revised manuscript, we have improved the transfection protocol, and the new results showed that Tyr signals were reduced significantly in both RUTBC1-expressing and ANKR1-expressing B16-F1 cells (Figure 4D and 4E).  We have included these new data and edited the text to avoid confusion (page 8).

  1. The authors described the distribution of Tyr and Tyrp1 (line266, p7). But Tyr and Tyrp1 seem to be more localized near the Golgi, like melan-cht/siRab32 cells in Figure 5E. Is it really no change of Tyr or Tyrp1 distribution in Rab32/38 B16-F1 cells?  If there is some quantification of Tyr or Tyrp1 distribution in WT and Rab32/28-depleted B16-F1 cells, it is better to be noted in text.  In addition, it is necessary to describe the relationship between the change of distribution of Tyr and melanogenesis, to clarify which distribution of Tyr is important for melanogenesis.

              We understand this comment.  Both Tyr and Tyrp1 are distributed to the “cell periphery” in addition to the Golgi in B16F1 cells and melanocytes.  As far as we have tested, no change in the distributions of Tyr and Tyrp1 was observed between WT and Rab32/38-KO cells.  To avoid a misunderstanding, we have replaced Rab32/38-KO cell images with more representative one (new Fig. 5E).  We have also described that Tyr and Tyrp1 were restricted to the perinuclear region (i.e., lack of peripheral signal) of melan-le cells, which exhibited a transparent phenotype (Figure 2D and page 4).

  1. in p2, line85, “insvestingating” should be investigating

              We apologize for this mistake.  We have corrected it in the revised manuscript.

  1. In Figure 1B, is it 10 nt deletion instead of “insertion”?

              We apologize for this mistake.  We have corrected Figure 1B in the revised manuscript.

  1. Figure 3 legend should be in line 190, but not 200 in p5.

             As suggested, we have corrected it in the revised manuscript.

  1. In Figure S1, “E” should be C.

              We apologize for this mistake.  We have corrected it in the revised manuscript.

Round 2

Reviewer 1 Report

My concerns were addressed.